# Control of Paneth cell function by HuR regulates gut mucosal growth by altering stem cell activity

Lan Xiao[1], Bridgette Warner[1], Caroline G Mallard[1], Hee K Chung[1], Amol Shetty[2], Christine A Brantner[3] , Jaladanki N Rao[1,4], Gregory S Yochum[5,6], Walter A Koltun[5], Kathleen B To[4], Douglas J Turner[1,4], Myriam Gorospe[7], Jian-Ying Wang[1,4,8] 

**Rapid self-renewal of the intestinal epithelium requires the activity of intestinal stem cells (ISCs) that are intermingled with Paneth cells (PCs) at the crypt base. PCs provide multiple secreted and surface-bound niche signals and play an important role in the regulation of ISC proliferation. Here, we show that control of PC function by RNA-binding protein HuR via mitochondria affects intestinal mucosal growth by altering ISC activity. Targeted deletion of HuR in mice disrupted PC gene expression profiles, reduced PC-derived niche factors, and impaired ISC function, leading to inhibited renewal of the intestinal epithelium. Human intestinal mucosa from patients with critical surgical disorders exhibited decreased levels of tissue HuR and PC/ISC niche dysfunction, along with disrupted mucosal growth. HuR deletion led to mitochondrial impairment by decreasing the levels of several mitochondrial-associated proteins including prohibitin 1 (PHB1) in the intestinal epithelium, whereas HuR enhanced PHB1 expression by preventing microRNA-195 binding to the *Phb1* mRNA. These results indicate that HuR is essential for maintaining the integrity of the PC/ISC niche and highlight a novel role for a defective PC/ISC niche in the pathogenesis of intestinal mucosa atrophy.**

## Introduction

The maintenance of homeostasis in the gut epithelium is a complex process that requires epithelial cells to rapidly alter gene expression patterns to regulate cell survival, proliferation, migration, differentiation, and cell-to-cell interaction (Xian et al, 2017; Borrelli et al, 2021; Yang et al, 2021). This homeostasis is disrupted in different situations, including surgical patients who undergo massive gastrointestinal surgical resection and are then supported with total parenteral nutrition. The disrupted renewal of the intestinal

epithelium in critically ill patients impairs mucosal adaptation and causes gut barrier dysfunction, which can then lead to the translocation of luminal toxic substances and bacteria to the bloodstream, sepsis, and, in some instances, multiple organ dysfunction syndrome and death (Carter et al, 2013; Kumar et al, 2020). Effective therapies to preserve the integrity of the intestinal epithelium in patients with critical surgical illnesses are limited, as the mechanisms that regulate gut mucosal renewal in stressful environments are poorly understood.

The dynamic turnover rate of the human small intestinal epithelium, which undergoes ~$10^{11}$ mitoses/day, is driven by intestinal stem cells (ISCs) and is extensively regulated (Radtke & Clevers, 2005; Serra et al, 2019). ISCs divide daily and produce bipotent progenitors, amplifying and differentiating into absorptive or secretory lineages (Sailaja et al, 2016; Li et al, 2023). Paneth cells (PCs), specialized intestinal epithelial cells (IECs) residing at the intestinal crypt base, produce abundant antibacterial proteins and peptides such as lysozyme and Reg3 lectins (Sato et al, 2011; Yu et al, 2020a). PCs also create a niche for ISCs in the crypts and provide multiple secreted and surface-bound niche signals that determine ISC fate (Sato et al, 2011; Yilmaz et al, 2012; Chung et al, 2021). Emerging evidence indicates that interactions between PCs and ISCs are crucial for constant intestinal epithelial renewal under various pathophysiological conditions (Rodriguez-Colman et al, 2017; Baulies et al, 2020; Butto et al, 2020; Lueschow & McElroy, 2020). Coculturing of sorted PCs with ISCs strongly enhances intestinal organoid formation and growth (Sato et al, 2011; Yu et al, 2020a), whereas PC defects in the crypts result in ISC dysfunction and inhibit intestinal mucosal growth (Shroyer et al, 2007; Norona et al, 2020). When PCs are ablated genetically in mice, intestinal enteroendocrine, tuft, and stromal cells can also produce niche factors and support ISCs (Durand et al, 2012; van Es et al, 2019).

HuR (encoded by the *Elavl1* gene), one of the most prominent RNA-binding proteins regulating mRNA translation and turnover, is widely involved in many aspects of gut mucosal pathobiology (Li et al, 2020; Palomo-Irigoyen et al, 2020). In our previous studies, we

[1]Cell Biology Group, Department of Surgery, University of Maryland School of Medicine, Baltimore, MD, USA  [2]Institute for Genome Science, University of Maryland School of Medicine, Baltimore, MD, USA  [3]Electron Microscopy Core Imaging Facility, University of Maryland Baltimore, Baltimore, MD, USA  [4]Baltimore Veterans Affairs Medical Center, Baltimore, MD, USA  [5]Department of Surgery, Pennsylvania State University College of Medicine, Hershey, PA, USA  [6]Department of Biochemistry and Molecular Biology, Pennsylvania State University College of Medicine, Hershey, PA, USA  [7]Laboratory of Genetics and Genomics, National Institute on Aging-IRP, NIH, Baltimore, MD, USA  [8]Department of Pathology, University of Maryland School of Medicine, Baltimore, MD, USA

Correspondence: jywang@som.umaryland.edu

found declines in the levels of tissue HuR - along with inhibition of the growth of the intestinal mucosa - in patients with various illnesses, including inflammatory bowel disease (IBD) (Xiao et al, 2019; Li et al, 2020). Specific ablation of HuR in the intestinal epithelium of mice (IE-HuR$^{-/-}$) causes atrophy of the mucosa in the small intestine and compromises the regeneration of the gut mucosa and adaptation after irradiation (Liu et al, 2014), septic stress (Zhang et al, 2020), and mesenteric ischemia and reperfusion (Liu et al, 2017). Recently, HuR was shown to regulate PC function in the intestinal epithelium by altering the membrane localization of Toll-like receptor 2 via posttranscriptional control of chaperone protein CNPY3 (Xiao et al, 2019; Chung et al, 2021). However, little is known about the role and mechanism of altered PCs by HuR, particularly their interaction with ISCs, in the regulation of intestinal mucosal growth.

Mitochondria generate ATP to provide cells with energy and function as factories that participate actively in the biosynthesis of various macromolecules (Urbauer et al, 2020). As unveiled by recent studies, by virtue of their ability to produce a plethora of metabolites, mitochondria in the gut epithelium appear to play an emerging role as signaling organelles (Twig et al, 2008; Sato et al, 2021). Mitochondria sense the metabolic environment and integrate host and microbial-derived signals (Berger et al, 2016; Ludikhuize et al, 2020). Mitochondrial metabolism, dynamics, and stress responses are pivotal in regulating intestinal mucosal self-renewal and epithelial differentiation (Berger et al, 2016; Jackson et al, 2020; Ludikhuize et al, 2020). Mitochondrial impairment delays gut mucosal repair and weakens epithelium–host defenses. Targeted deletion of the genes encoding mitochondrial proteins prohibitin 1 (PHB1) and heat shock protein 60 (HSP60) in mice results in PC defects and causes a predisposition to ileitis in mice (Jackson et al, 2020; Khaloian et al, 2020). Mitochondrial activity is attenuated in human colonic mucosal tissues from patients with ulcerative colitis (Ho & Theiss, 2022; Ozsoy et al, 2022).

In this study, we provide evidence that control of PC function by HuR regulates the renewal of the small intestinal epithelium by affecting ISC activity. We found that loss of HuR in mice altered cell type-specific gene expression in PCs and led to dysfunction of the PC/ISC niche. Human intestinal mucosa from patients with critical surgical disorders exhibited both reduced HuR and defects in PC/ISC niche function, which was associated with an inhibition of intestinal epithelial renewal. Our results further show that HuR enhances mitochondrial function by increasing the levels of mitochondrial-associated proteins including PHB1, and that it promotes PHB1 expression levels at least in part by interfering with the function of microRNA-195 (miR-195). These findings reveal that HuR is essential for sustaining integrity of the PC/ISC niche in the intestinal epithelium and point to the HuR/miR-195/PHB1 axis as novel therapeutic targets for interventions to enhance the function of the PC/ISC niche and promote intestinal mucosal regeneration and adaptation in patients with critical disorders.

# Results

## HuR deletion causes defects in the PC/ISC niche function in the mucosa of the small intestine

To determine if HuR promotes intestinal epithelial renewal by augmenting ISC proliferation via PCs, we used IE-HuR$^{-/-}$ mice that

were generated by crossing HuR$^{fl/fl}$ mice with villin-Cre–expressing mice as described (Liu et al, 2014). HuR levels in the mucosa of the small intestine and colon were undetectable in IE-HuR$^{-/-}$ mice, but there were no changes in HuR expression in other tissues and organs such as the gastric mucosa, lung, liver, kidney, and pancreas, as reported previously (Liu et al, 2014, 2017). Immunohistochemistry analysis showed that HuR staining almost completely disappeared in epithelial cells in the small intestinal mucosa of IE-HuR$^{-/-}$ mice, although HuR expression levels were unaffected in submucosal connective tissue (Fig 1A). Conditional HuR deletion in IECs inhibited the renewal of the small intestinal epithelium, as indicated by a decrease in the levels of BrdU-positive cells within the crypts and subsequent shrinkages of crypt and villi, but it failed to affect the growth of the colonic mucosa - similar to the observations reported in our previous study (Liu et al, 2014, 2017). Targeted HuR deletion in mice did not alter the overall morphology or structure of the mucosa of the small intestine or the colon. HuR deletion in the intestinal epithelium led to abnormalities in PCs, as evidenced by decreased lysozyme-positive cells in IE-HuR$^{-/-}$ mice relative to control littermate mice (Fig 1B), as observed previously (Xiao et al, 2019). Interestingly, defects in PCs induced by HuR deletion were associated with a loss of ISCs in the small intestinal mucosa. Staining of whole mounts of the small intestine revealed that ISCs, marked by OLFM4 and LGR5, were normally located at the base of the crypts in littermate mice, but the numbers of OLFM4- and LGR5-positive cells (Fig 1C and D) decreased markedly in IE-HuR$^{-/-}$ mice when compared with control littermates. On the other hand, HuR deletion did not affect the number of goblet cells and differentiated enterocytes, as measured by Alcian blue staining and villin immunostaining analysis, respectively. To examine how HuR affected the interaction of PCs with ISCs in the small intestinal mucosa after HuR deletion, we examined changes in the levels of niche signals derived from PCs in IE-HuR$^{-/-}$ and littermate mice. As shown in Fig 1E, immunostaining detection of WNT3 and NOTCH2 revealed enriched signals at the base of small intestinal crypts in control littermate mice, but their levels in intestinal crypts decreased markedly in IE-HuR$^{-/-}$ mice. Semiquantitative analysis showed that the intensities of WNT3 and NOTCH2 immunostaining decreased by >90% in the HuR-deficient intestinal epithelium compared with those observed in littermate mice (Fig 1F). Because WNT3 and NOTCH2 are niche growth factors (Xian et al, 2017; Yu et al, 2018; Bottcher et al, 2021), these results suggest that HuR deletion impairs ISC proliferation at least partially by reducing PC-derived niche signals.

## Altered cell type-specific gene expression profiles in the HuR-deficient intestinal epithelium

To obtain cell type-specific transcriptional regulation profiles in the intestinal epithelium after HuR deletion, we performed single-cell RNA-sequencing (scRNA-seq) analysis on the small intestinal mucosa harvested from IE-HuR$^{-/-}$ and control littermate mice. In total, 60,000 high-quality cells, as characterized (Fig S1A) and validated by flow-cytometry analysis (Fig S1B–D), were grouped into 18 clusters, which expressed classic cell type markers from the absorptive or secretory lineages, and markers distinctive of ISCs and daughter transient-amplifying cells (Fig S2), following

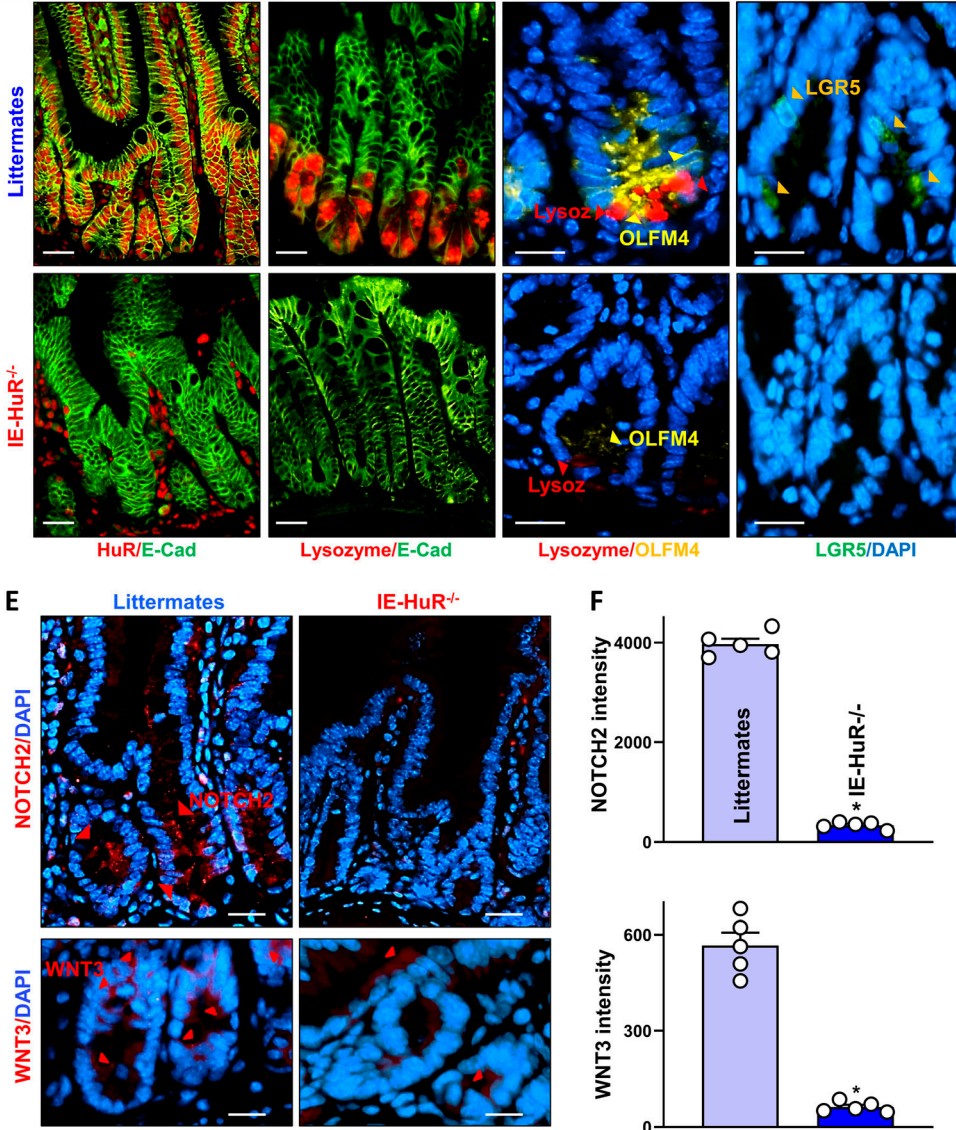

**Figure 1. Targeted deletion of HuR in mice causes defects in the PC/intestinal stem cell niche in the intestinal epithelium.**
**(A)** Immunostaining of HuR in the small intestinal mucosa of IE-HuR$^{-/-}$ and control littermate mice. Red, HuR; and green, E-cadherin (E-cad). Scale bars, 50 $\mu$m. Three separate experiments showed similar results. **(B)** Images of Paneth cells marked by lysozymes in the small intestinal mucosa. Red, lysozyme; and green, E-cad. Scale bars, 50 $\mu$m. **(C)** Immunostaining of OLFM4 in the small intestinal crypts. Yellow, OLFM4; blue, nucleus stained by DAPI. Scale bar, 25 $\mu$m. **(D)** Immunostaining of LGR5 in the small intestinal crypts, as shown in yellow–green. Scale bar, 25 $\mu$m. **(E)** Immunostaining of NOTCH2 (*top*) and WNT3 (*bottom*) in the small intestinal crypts, as shown in red. Scale bar, 25 $\mu$m. **(F)** Quantitation of NOTCH2 (*top*) and WNT3 (*bottom*) signals in the small intestinal crypts treated as described in (E). Values are means ± SEM (*n* = 5 biological replicates). Unpaired, two-tailed *t* test was used. *P < 0.05 compared with control littermates.

published single-cell transcriptomic signatures in the intestinal epithelium (Islam et al, 2020; Elmentaite et al, 2021). Secretory cells in the epithelial compartment of the small intestine consisted of PCs, goblet cells, tuft cells, and enteroendocrine cells in both IE-HuR$^{-/-}$ and littermate mice, whereas absorptive cells were marked by various enterocytes (Fig 2A). Notably, PCs were marked by high expression of *Defe5* and *Rg3a* mRNAs, whereas ISCs were marked by transcripts including *Ascl2*, *Lgr5*, *Olfm4*, *Rgmb*, and *Smoc2* mRNAs, as reported previously (Elmentaite et al, 2021; Luna Velez et al, 2023). Cell cluster analysis revealed that the numbers of PCs and ISCs decreased in IE-HuR$^{-/-}$ mice compared with control littermate mice, which were consistent with the findings obtained from studies using immunostaining assays as shown in Fig 1.

Profiling analysis of cell type-specific gene expression showed the existence of transcriptionally distinct PCs and ISCs in the small intestinal epithelium of IE-HuR$^{-/-}$ mice. A comparison of the

transcriptomic profiles in the mucosa from IE-HuR$^{-/-}$ mice relative to control littermates demonstrated that ~2,116 RNAs were differentially abundant in PCs; 1,144 were less abundant and 972 were more abundant after HuR deletion (Fig 2B, *top*). In ISCs, ~400 RNAs were differentially abundant - 289 RNAs were lower, and 120 RNAs were higher in IE-HuR$^{-/-}$ mice compared with control littermates (Fig 2C, *top*). The most highly increased and decreased RNAs in PCs and ISCs of IE-HuR$^{-/-}$ mouse relative to control littermates are summarized in low panels of Fig 2B and C. Amongst the top differentially abundant mRNAs are many encoding niche growth factors and proteins that are necessary for maintaining the PC structure and function. HuR deletion also altered gene expression profiles of secretory progenitor cells (Atoh1+) and goblet cells in the small intestinal epithelium, but the exact role of HuR in the regulation of secretory progenitor and goblet cells will be fully investigated in a separate study. Collectively, our single-cell sequencing data analysis identified transcriptomic and cellular

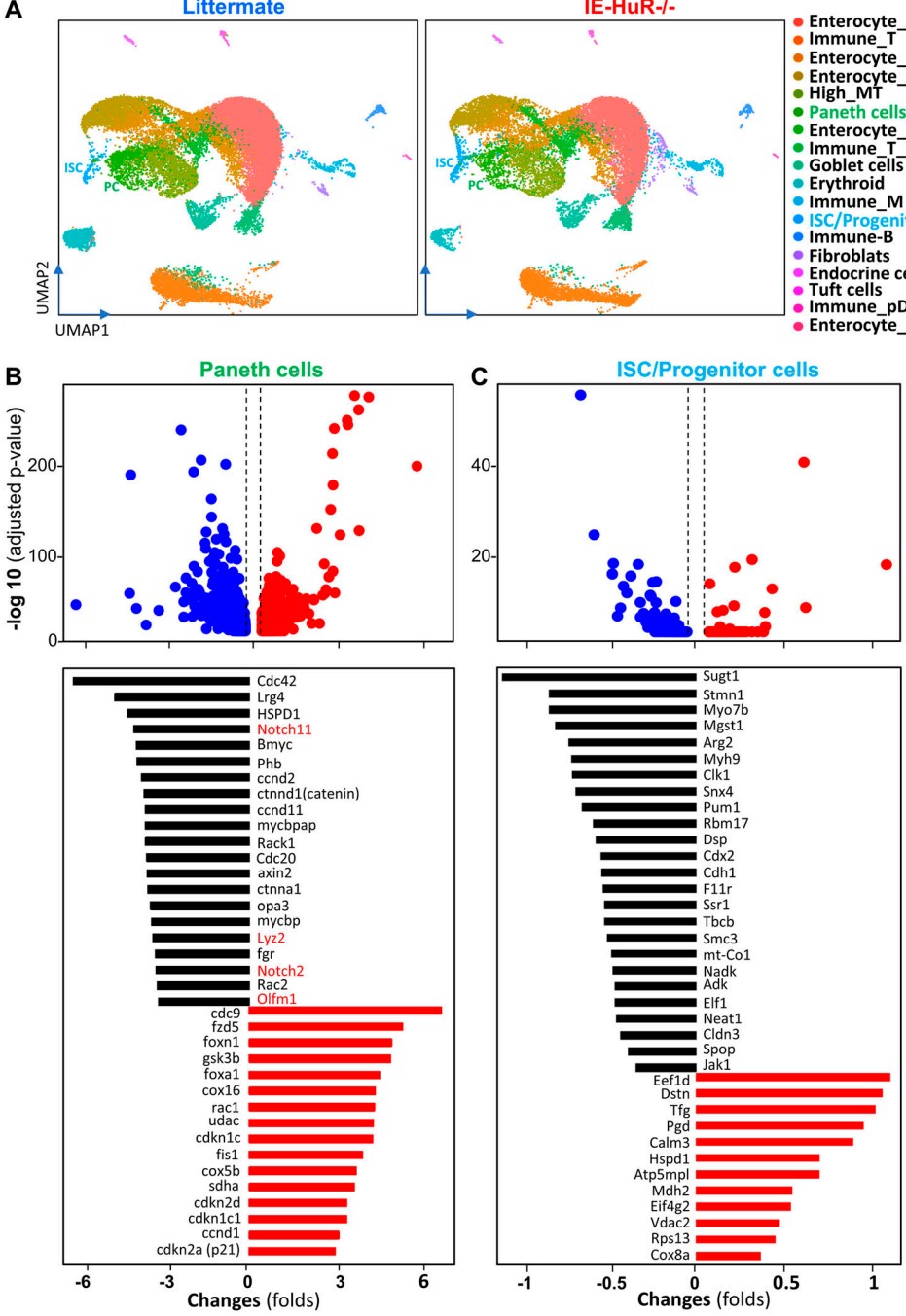

**Figure 2. Single-cell transcriptional profiles of PCs and intestinal stem cells (ISCs) in the small intestinal epithelium after HuR deletion in mice.**
**(A)** Uniform manifold approximation and projection of key epithelial cell types of the small intestinal mucosa from control littermates (*left*) and IE-HuR$^{-/-}$ mice (*right*) (*n* = 3). Annotated clusters of PCs are showed in green; ISCs are shown in light-blue. **(B)** *Top panel*, scatter plot depictions of genes expressed differentially in PCs of control and IE-HuR$^{-/-}$ mice, as measured by scRNA-seq analysis. *Low panel*, differential expression analysis of most altered genes in results described in *top panel*. Values are the means from three animals. The *P*-value cutoff used for identifying differentially expressed genes was 0.05. **(C)** *Top panel*, scatter plot depictions of genes expressed differentially in ISCs. *Low panel*, most altered genes in ISCs in results described in the *top panel*. Values are the means from three animals. The *P*-value cutoff used for identifying differentially expressed genes is 0.05.

responses to HuR deletion in the small intestinal mucosa and strongly suggested the importance of HuR in the PC/ISC niche to regulate the renewal of the intestinal epithelium.

## Reduced HuR levels associate with abnormal PC/ISC niche in patients with critical illness

To explore the clinical relevance of HuR function in the human PC/ISC niche, human ileal mucosal tissues were collected from four Crohn's disease (CD) patients who required urgent/emergent

intestinal resection because of severe complications (SC) such as intestinal perforation, peritonitis, and necrotizing enteritis as well as four healthy controls who had neither CD, nor emergency surgical disorders. Consistent with findings observed in IBD patients without SC (Xiao et al, 2019), the ileal mucosa from patients with CD/SC also exhibited a significant decrease in the levels of HuR (Fig 3A), as measured by immunostaining analysis. In the mucosa from the small intestine, HuR was located in both the cytoplasm and nucleus in control individuals, but the levels of total HuR and cytoplasmic HuR were significantly lower in CD/SC patients relative to control patients.

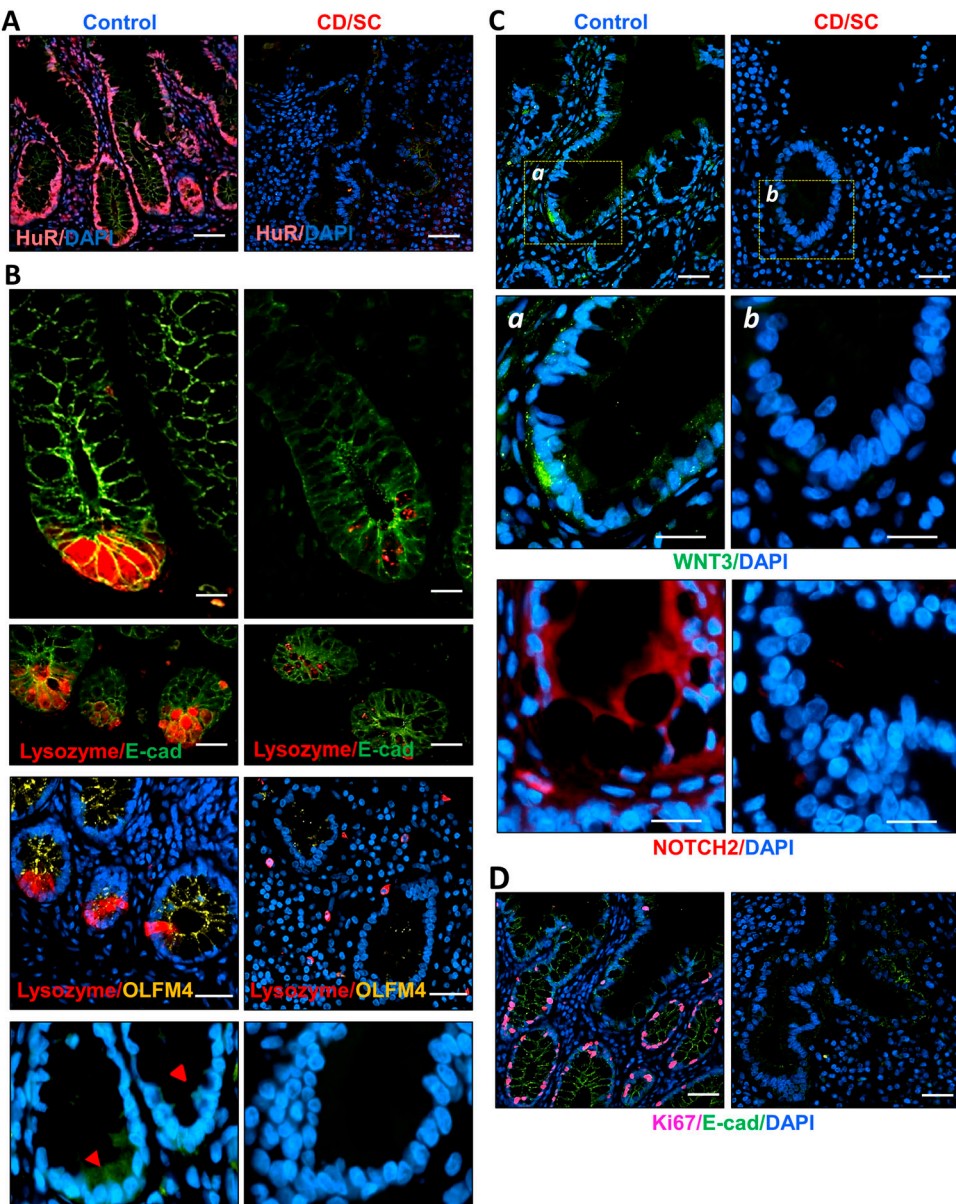

**Figure 3. Reduced HuR associates with defects in PC/intestinal stem cell niches in patients with CD/SC.**
**(A)** Immunostaining of HuR in the small intestinal mucosa from control individuals and Crohn's disease patients who underwent emergency surgery because of severe complications (CD/SC). Purple, HuR; blue, nucleus stained by DAPI. Scale bars, 50 μm. All these experiments were repeated in four controls and four patients with CD/SC and showed similar results. **(B)** Images of PCs (lysozyme-positive cells; *top*) and intestinal stem cells marked by OLFM4 and LGR5 (*bottom*) in the human small intestinal mucosa as examined by immunostaining assays using anti-lysozyme (red), anti-OLFM4, anti-LGR5 (yelloe-green), and anti-E-cadherin (green). Blue, nucleus stained by DAPI. Scale bars, 50 and 25 μm. **(C)** Immunostaining of WNT3 (*top*) and NOTCH2 (*bottom*) in the crypts of human small intestine. Yellow–green, WNT3; red, NOTCH2; and blue, nucleus stained by DAPI. Scale bar, 25 μm. **(D)** Small intestinal mucosal renewal in patients with CD/SC, as assessed by measuring Ki67 staining (pink). Scale bar, 50 μm.

Importantly, reduced levels of HuR in the intestinal mucosa in patients with CD/SC were associated with significant defects in PC/ISC niche function because the numbers of lysozyme-positive cells and OLFM4- and LGR5-positive cells in the ileal mucosa from CD/SC patients decreased markedly relative to those observed in control patients (Fig 3B). In fact, PCs and ISCs were almost totally undetectable in some ileal mucosal tissue samples from CD/SC patients. Consistent with the observations in the small intestinal mucosa of IE-HuR$^{-/-}$ mouse (Fig 1), decreased HuR levels in the mucosa of patients with CD/SC were accompanied by reduced levels of PC-derived niche signals, as evidenced by decreased intensities of WNT3 and NOTCH2 immunostaining signals in the crypts of CD/SC patients compared with control individuals (Fig 3C). Importantly,

the impaired function of the PC/ISC niche associated with HuR inhibition in patients with CD/SC was linked to a significant inhibition of the intestinal mucosal growth because the levels of the proliferation marker Ki67 decreased markedly in CD/SC patients relative to control individuals (Fig 3D). These results implicate PC/ISC niche dysfunction in the impaired renewal of the intestinal epithelium in critically ill patients after decreased tissue HuR levels.

## HuR deletion impairs mitochondrial function in the intestinal epithelium

Because mitochondria play an important role in maintaining intestinal epithelial homeostasis and its dysfunction is intimately

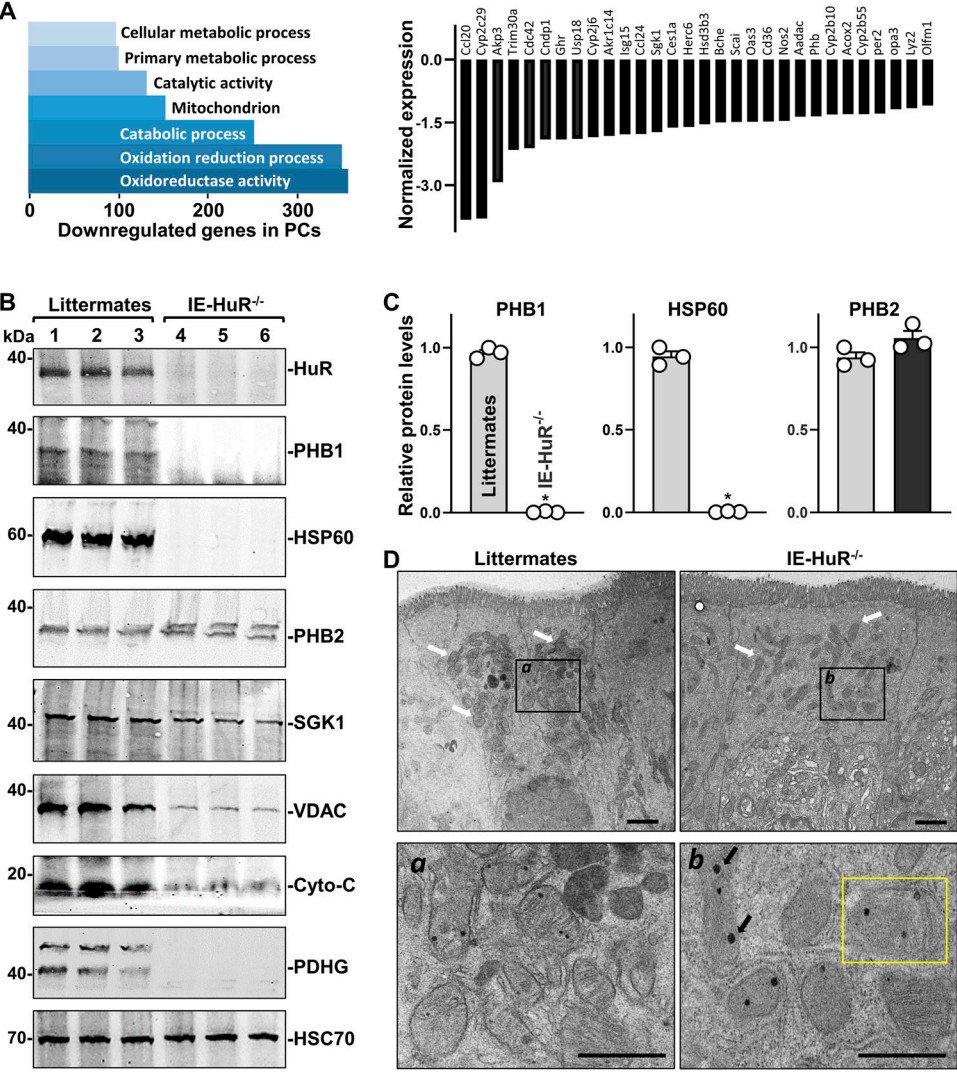

**Figure 4. HuR deletion in mice inhibits the expression of mitochondrial proteins in the intestinal epithelium.**
**(A)** Causal analysis of metabolism/mitochondria-associated pathways down- or up-regulated in PCs after HuR deletion in mice. *Left panel*, most altered pathways involved in metabolism and mitochondrial activity in HuR-deficient PCs. *Right panel*, most altered genes in results described in *left panel*. Values are the means from three animals. The *P*-value cutoff used for identifying differentially expressed genes was 0.05. **(B)** Immunoblots of HuR and various mitochondrial proteins in the small intestinal mucosa of littermate and IE-HuR$^{-/-}$ mice. **(C)** Quantitative analysis derived from densitometric scans of immunoblots of mitochondrial proteins in results as described in (B). Values are the means ± SEM (*n* = 3). Unpaired, two-tailed *t* test was used. *$P < 0.05$ compared with control littermates. **(D)** Transmission electron microscopy of crypt bases of the small intestinal epithelium. White arrows: mitochondria; black arrows: dense inclusion body; yellow box: unhealthy mitochondria. Scale bar = 2 and 0.5 $\mu$m. Three separate experiments showed similar results.
Source data are available for this figure.

involved in the pathogenesis of IBD and other mucosal pathologies (Berger et al, 2016; Ludikhuize et al, 2020; Urbauer et al, 2020), we tested the possibility that HuR regulates activity of the PC/ISC niche by affecting mitochondrial metabolism. We further screened for expression of mitochondrial-associated genes in our single-cell sequencing data from PCs and found that HuR deletion decreased the expression levels of many mRNAs encoding proteins that control mitochondrial catabolism and oxidation/reduction reactions (Fig 4A). To further evaluate if IE-HuR$^{-/-}$ mice had mitochondrial abnormalities, we examined changes in expression of several mitochondrial-associated proteins in the small intestinal mucosa. HuR ablation in mice decreased the levels of PHB1 and HSP60 (Fig 4B and C); these two proteins are essential for mitochondrial integrity in the intestinal epithelium and loss of PHB1 or HSP60 results in defective PCs and leads to ileitis in mice (Jackson et al, 2020; Khaloian et al, 2020). PHB1 and HSP60 were highly expressed in the small intestine mucosa of control littermate mice, but their levels decreased markedly in IE-HuR$^{-/-}$ mice. In fact, PHB1 and HSP60 proteins were undetectable in the small intestinal mucosa of IE-HuR$^{-/-}$ mice by Western blot analysis (Fig 4B, top).

In addition, the HuR-deficient mucosa of the intestinal mucosa also exhibited decreased levels of VDAC, cytochrome C (Cyto-C), and PDHG without changes in the abundance of PHB2, SGK or HSC70 (Fig 4B, *bottom*; Fig S3A). Moreover, transmission electron microscopy (TEM) at the base of crypts in the small intestinal epithelium showed reduced numbers of mitochondria in PC-like cells from IE-HuR$^{-/-}$ mice, and mitochondria with swollen morphology, disruption of cristae, decreased fused structures, and occasional dense inclusion bodies, when compared with littermate mice (Fig 4D). On the other hand, HuR deletion did not alter the levels or morphology of mitochondria in enterocytes located at the villous area of the small intestinal mucosa, where all mitochondrial features were indistinguishable between IE-HuR$^{-/-}$ mice and littermate mice, as examined by TEM (Fig S3B).

To analyze mitochondrial function in the HuR-deficient intestinal epithelium, we analyzed primary cultured intestinal organoids and cultured IECs. Consistent with the observations in the mouse epithelium, HuR deletion inhibited the growth of the intestinal organoids ex vivo, with a marked decrease in the numbers of

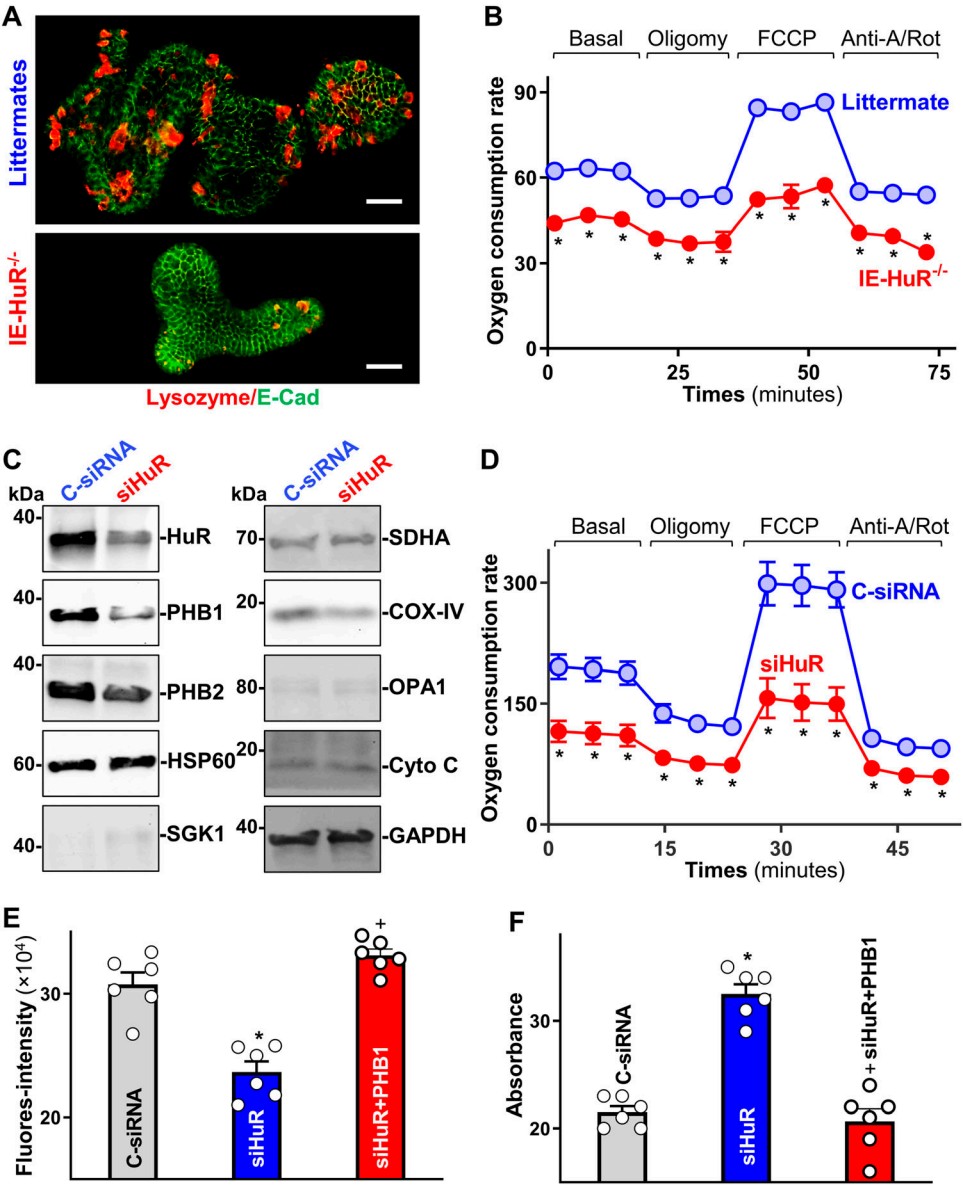

**Figure 5. Decreasing the levels of HuR causes mitochondrial dysfunction.**
**(A)** Immunostaining of lysozyme-positive cells in the small intestinal organoids derived from control littermate and IE-HuR$^{-/-}$ mice. Red, lysozyme; green, E-cadherin. Scale bars, 80 $\mu$m. **(B)** Mitochondrial respiration in the intestinal organoids treated as described in (A), as measured by Seahorse analysis. Values are the means ± SEM ($n$ = 3). Two-way ANOVA with Bonferroni post hos test was used. *$P$ < 0.05 compared with control littermates. **(C)** Immunoblots of HuR and mitochondrial proteins in cultured intestinal epithelial cells (IECs) transfected with siRNA directed at silencing HuR (siHuR) or control siRNA (C-siRNA). The levels of the proteins shown were examined 48 h after the transfection. **(D)** Mitochondrial respiration in cultured IECs treated as described in (C). Values are the means ± SEM ($n$ = 3). Two-way ANOVA with Bonferroni post hos test was used. *$P$ < 0.05 compared with C-siRNA. **(E, F)** Levels of MitoTracker Green and superoxide in cultured IECs after transfection with siHuR alone or co-transfection with siHuR and prohibitin 1 expression vector ($n$ = 6). *, +$P$ < 0.05 compared with C-siRNA and siHuR, respectively.

lysozyme-positive cells in intestinal organoids derived from IE-HuR$^{-/-}$ mice and smaller organoids that contained fewer buds when derived from HuR-deficient mice compared with those generated from control littermate mice (Fig 5A). To examine the mitochondrial respiratory capacity, we used a Cell Mito Stress test and a Seahorse XF extracellular flux analyzer. We observed decreases in basal and maximal respiration levels and in ATP production, and decreased spare respiratory capacity in the organoids derived from IE-HuR$^{-/-}$ mice compared with the organoids generated from control littermate mice (Fig 5B). Using cultured human colorectal adenocarcinoma Caco-2 cells, we found that HuR silencing similarly disrupted mitochondrial homeostasis and reduced the levels of several mitochondrial proteins. Decreasing HuR by transfection with a specific siRNA targeting HuR (siHuR) markedly decreased the levels of PHB1 and COX-IV, although it only slightly

reduced PHB2 abundance and failed to alter the expression of HSP60, SDHA or Cyto-C (Figs 5C and S4A). Basal levels of SGK1 and OPA1 proteins in Caco-2 cells were undetectable. On the other hand, ectopic overexpression of HuR by transfection with an HuR expression vector significantly increased the levels of cellular PHB1 and modestly elevated COX-IV levels (Fig S4B). Seahorse analysis showed similar inhibitory patterns of mitochondrial respiratory capacity and ATP production in HuR-silenced Caco-2 cells (Fig 5D), as observed in HuR-deficient intestinal organoids (Fig 5B).

The impairment of mitochondrial function induced by HuR silencing was further confirmed by a decrease in the levels of MitoTracker green (Fig 5E) and an increase in superoxide production (Fig 5F), as examined by using MitoTracker Red and MitoSox kits available commercially. Interestingly, mitochondrial dysfunction in HuR-silenced cells was prevented by ectopically overexpressing PHB1,

as indicated by an increased level of MitoTracker Green and reduction in superoxide level when HuR-silenced cells were transfected with a vector to overexpress PHB1 (Fig 5E and F, right). Together, these results suggest that lower HuR levels trigger mitochondrial dysfunction at least in part by inhibiting PHB1 expression. We propose that these effects contribute to the impairment of the PC/ISC niche and subsequent mucosal atrophy.

### HuR regulates PHB1 expression by interfering with miR-195 function

Given that PHB1 is required for maintaining mitochondrial homeostasis and PC function as shown in the present study and previous reports (Jackson et al, 2020; Liu et al, 2022), we investigated the mechanism by which HuR regulates PHB1 expression in cultured IECs. First, we examined if HuR associated with the *Phb1* mRNA by performing RNP immunoprecipitation (IP) assays using an anti-HuR antibody under conditions that preserved RNP integrity (Yu et al, 2011). The interaction of *Phb1* mRNA with HuR was examined by isolating RNA from the IP material and subjecting it RT, followed by quantitative (Q)–PCR analysis. Although HuR overexpression increased PHB1 levels (Fig S4B) and HuR silencing decreased *Phb1* mRNA levels (Fig 6A), HuR did not specifically bind to *Phb1* mRNA, as the levels of *Phb1* mRNA in HuR IP samples were similar to those observed in control IgG (Fig 6B). As a positive control, the *claudin 1* (*Cldn1*) mRNA was highly enriched in HuR samples compared with control IgG. These results indicate that HuR may regulate PHB1 expression through mechanisms other than by directly interacting with the *Phb1* mRNA.

Second, we focused on the role of microRNA miR-195 (miR-195) because miR-195 represses PHB1 production (Cirilo et al, 2017) and because HuR and miR-195 jointly regulate expression of shared target transcripts antagonistically (Zhuang et al, 2013; Kwon et al, 2021). As shown by pulldown analysis after transfecting Caco-2 cells with a biotinylated miR-195 (Kwon et al, 2021), miR-195 directly interacted with the *Phb1* mRNA in Caco-2 cells but not with *Phb2* mRNA (Fig 6C and D); transfections with a control biotinylated scramble RNA did not show enrichment in *Phb1* mRNA. Ectopically expressing miR-195 by transfecting the miR-195 precursor (pre-miR-195) (Fig 6E) markedly decreased the levels of cellular *Phb1* mRNA (Fig 6F, top) and protein (Fig 6F, bottom; Fig S5A), but it only slightly reduced PHB2 and failed to alter the abundance of HSP60, COX-IV, and Cyto-C proteins. Interestingly, increasing the levels of cellular HuR by transfection with the HuR expression vector reduced the ability of biotinylated miR-195 to bind to the *Phb1* mRNA, relative to what was seen in the vector control group (Fig 6G).

Furthermore, HuR-induced stimulation of PHB1 expression was prevented by increasing miR-195 through transfection with pre-miR-195 (Figs 6H and S5B). The levels of PHB1 protein in cells co-transfected with HuR and pre-miR-195 were similar to those observed in cells transfected with control vector. In addition, there were no significant differences in the levels of PHB2 between these three groups. These results indicate that HuR promotes PHB1 expression primarily by inhibiting the association of *Phb1* mRNA with miR-195.

## Discussion

Integrity and effectiveness of the PC/ISC niche are essential for constant renewal of the intestinal epithelium (Sato et al, 2011; Yilmaz et al, 2012; Butto et al, 2020), but the exact mechanism underlying the activation of PC/ISC niche in response to stress remains largely unknown. In the present study, we provide genetic evidence that HuR plays an important role in the PC/ISC niche function at least in part by controlling mitochondrial activity. Targeted deletion of HuR in mice caused deregulation of cell type-specific gene expression in PCs, resulted in defective PCs, and impaired ISC proliferation, in turn inhibiting the growth of the small intestine mucosa. Experiments aimed at characterizing HuR targets in this process revealed that HuR deletion decreased mitochondrial metabolism by reducing the levels of expression of several mitochondrial-associated proteins including PHB1, and that HuR enhanced PHB1 expression by preventing miR-195 binding to the *Phb1* mRNA. These findings link HuR-regulated functions in the PC/ISC niche with intestinal epithelium renewal and highlight the connections between a dysfunctional PC/ISC niche, HuR deficiency, and intestinal epithelium pathology in patients with critical disorders.

The small intestinal epithelium is the fastest self-renewing tissue in mammals and this continuous growth is carried out by active ISCs, which reside at the base of the crypts and are intermingled with postmitotic and differentiated PCs (Sato et al, 2011; Sailaja et al, 2016). PCs constitute the ISC niche, secrete stem cell growth signals, and are thus essential for maintaining ISC function (Sato et al, 2011; Tian et al, 2015; Lueschow & McElroy, 2020). This dependence of ISCs on PC-mediated paracrine signaling can be easily recapitulated in an ex vivo system. For example, single ISCs can be expanded ex vivo into epithelial organoids or "mini-gut," but the outgrowth efficiency of single ISCs is quite low; when ISCs are plated together with PCs, however, their outgrowth efficiency increased by 10fold (Sato et al, 2011; Dayton & Clevers, 2017). In addition, PCs also metabolically support ISCs by providing them with a metabolic fuel source (Yu et al, 2020a). In the present study, we observed that HuR regulates ISC proliferation in the intestinal epithelium at least partially by altering PC function. Conditional deletion of HuR in mice altered the transcriptomic profiles of PCs and decreased the levels of PC-derived growth factors WNT3 and NOTCH2 in the crypt bases. Because ISCs are located at a niche growth factor-rich environment that relies on constant secretion of PCs, decreased levels of WNT3 and NOTCH2 in the HuR-deficient epithelium definitely contribute to ISC dysfunction and subsequent mucosal growth inhibition observed in IE-HuR$^{-/-}$ mice. In support of these results, HuR deletion in mice also decreases the expression of WNT co-receptor LRP6 at the posttranscription level in the intestinal epithelium (Liu et al, 2014).

The results presented here also show that HuR regulates PC function by controlling mitochondrial metabolism. Targeted HuR deletion resulted in mitochondrial dysfunction, as evidenced by decreased levels of mitochondrial-associated proteins in the intestinal epithelium and by an inhibition of mitochondrial respiratory capacity ex vivo and in vitro. PCs are highly susceptible to mitochondrial dysfunction driven by HuR deletion because cell

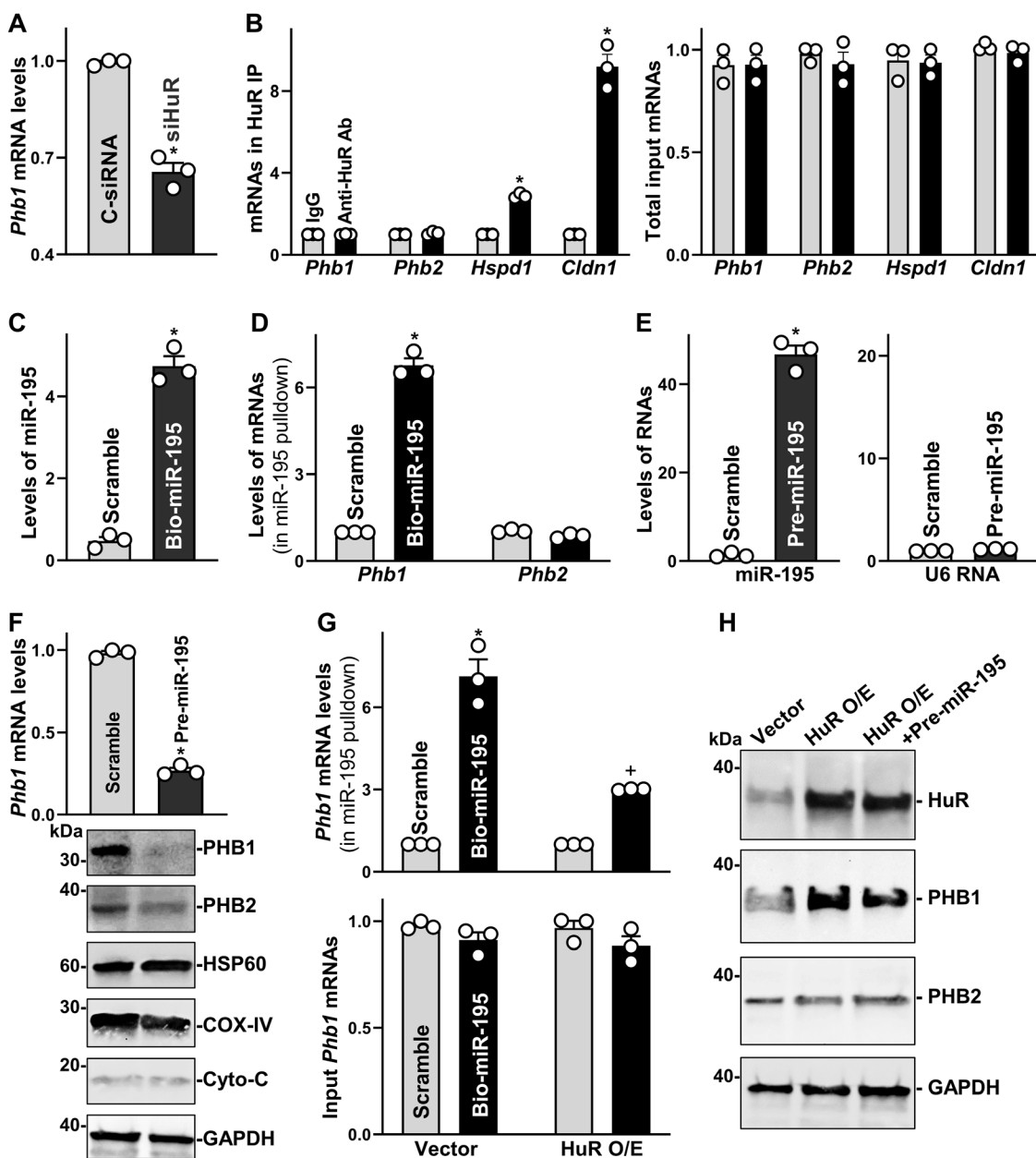

**Figure 6. HuR regulates prohibitin 1 (PHB1) expression via interaction with miR-195.**
**(A)** Levels of *Phb1* mRNA in Caco-2 cells 48 h after transfection with C-siRNA or siHuR. Values are the means ± SEM (*n* = 3 biological replicates). Unpaired, two-tailed *t* test was used. *P < 0.05 compared with C-siRNA. **(B)** *Left panel*, association of endogenous HuR with endogenous mRNAs in Caco-2 cells as measured by RIP using either anti-HuR antibody (Ab) or control IgG, followed by RT–PCR analysis. *Right panel*, levels of input mRNAs. Values are the means ± SEM (*n* = 3 biological replicates). Unpaired, two-tailed *t* test was used. *P < 0.05 compared with control IgG. **(C)** Levels of miR-195 in Caco-2 cells 24 h after transfection with biotinylated miR-195. Values are the means ± SEM (*n* = 3 biological replicates). Unpaired, two-tailed *t* test was used. *P < 0.05 compared with control scramble oligomer. **(D)** Levels of *Phb1* mRNA in the materials pulled down by biotin-miR-195 in cells treated as described in (C). Values are the means ± SEM (*n* = 3). Unpaired, two-tailed *t* test was used. *P < 0.05 compared with control scramble oligomer. **(E)** Levels of miR-195 (*left*) and *U6* RNA (*right*) in Caco-2 cells 48 h after transfection with pre-miR-195. Values are the means ± SEM (*n* = 3 biological replicates). Unpaired, two-tailed *t* test was used. *P < 0.05 compared with scramble. **(F)** Levels of *Phb1* mRNA (*top*) and protein (*bottom*) in cells treated as described in (E). Values are the means ± SEM (*n* = 3 biological replicates). Unpaired, two-tailed *t* test was used. *P < 0.05 compared with scramble. **(G)** Levels of *Phb1* mRNA in the materials pulled down by miR-195 (*top panel*) and total input mRNA (*bottom*) after co-transfection with HuR expression vector and bio-miR-195. Values are the means ± SEM (*n* = 3 biological replicates). *, + P < 0.05 compared with cells transfected with scramble oligomer or cells transfected with bio-miR-195 with control vector, respectively.
**(H)** Immunoblots of HuR and PHB1 proteins in Caco-2 cells 48 h after transfection with HuR expression vector alone or co-transfection with HuR vector and pre-miR-195. Experiments were repeated three times and showed similar results.
Source data are available for this figure.

type-specific gene expression analysis revealed the existence of transcriptionally distinct PCs in IE-HuR$^{-/-}$ mice and because HuR deletion caused ultrastructural abnormalities in the mitochondria of PC-like cells located at the bases of crypts. It has been reported that targeted deletion of the *Phb1* or *Hsp60* gene in IECs causes defects in PCs primarily by disrupting mitochondrial function, which compromises the epithelium–host defense and is the central to the pathogenesis of ileitis (Jackson et al, 2020; Khaloian et al, 2020). Interestingly, HuR deletion almost completely inhibited the expression of PHB1 and HSP60 in the intestinal epithelium, whereas ectopically expressed PHB1 restored mitochondrial function in HuR-deficient cells. Like other secretory cells such as goblet and tuft cells in the intestinal epithelium, PCs are mitochondrial-rich to sustain energy-expending secretion and other function; therefore, maintaining the mitochondrial health and effectiveness in PCs are especially crucial for their activity (Rath et al, 2018). In addition, mitochondrial impairment is deleterious in terminally differentiated long-lived cells such as PCs, because damaged mitochondria are not diluted or repaired by cell replication (Clevers & Bevins, 2013).

Another important finding of this study is that HuR regulates PHB1 expression by interacting with miR-195. Although HuR deletion decreased the levels of PHB1 in mouse intestinal mucosa and ectopic overexpression of HuR increased PHB1 levels in cultured cells, HuR failed to directly bind to the *Phb1* mRNA. In contrast, miR-195 was found to associate extensively with the *Phb1* transcript and inhibited PHB1 expression. Our study further shows that increasing the levels of cellular HuR blocked miR-195 association with the *Phb1* mRNA and that miR-195 overexpression abolished HuR-induced stimulation of PHB1 expression. As reported previously (Xiao & Wang, 2014; Wang et al, 2017; Ma et al, 2023), HuR can perform its regulatory function by antagonizing miRNAs and small ncRNAs besides its RNA binding affinity. For example, HuR antagonizes miR-548c-3p to regulate the expression of TOP2A (Srikantan et al, 2011), prevents miR-122-mediated repression of CAT-1 expression (Bhattacharyya et al, 2006), blocks miR-494 to regulate the expression of nucleolin (Tominaga et al, 2011), and co-operates with let-7 in repressing c-Myc expression (Kim et al, 2009). Although the exact process by which HuR competes with miR-195 for association with the *Phb1* mRNA remains unknown at present, HuR formed a complex with miR-195, thus abolishing miR-195 binding affinity for *Phb1* mRNA and preventing miR-195-induced inhibition of PHB1 expression. In this regard, HuR regulates stability of the *Stim1*, *Dclk1*, and *Cldn2* mRNAs by competing with miR-195 to bind to these transcripts, thus contributing to HuR-enhanced homeostasis of the intestinal epithelium (Zhuang et al, 2013; Kwon et al, 2021). Another possibility is that HuR may affect the processing of pre-miR-195, but our previous studies revealed that ectopically expressed HuR did not alter the levels of mature miR-195 (Kwon et al, 2021).

Our results are of particular importance from a clinical point of view, because human intestinal mucosa from patients with CD/SC exhibited both decreased levels of HuR and PC/ISC niche dysfunction. Although defective PCs and mitochondrial impairment are commonly observed in patients with IBD (Xiao et al, 2019; Ozsoy et al, 2022), the present study demonstrates for the first time that there were significant defects in PC/ISC niche in the intestinal mucosa of patients with critical surgical disorders after HuR inhibition. Notably, abnormalities in PC/ISC niche function in the

intestinal epithelium of IE-HuR$^{-/-}$ mouse are similar to those observed in the mucosa of patients with CD/SC. Our study also establishes a cause–effect relationship between HuR and control of PC/ISC niche function via mitochondria in in vivo, ex vivo, and cell culture models. Although most mechanistic studies were conducted in cultured Caco-2 cells and these in vitro observations should be verified in ex vivo and in vivo systems, our findings strongly support a model whereby HuR functionally interacts with miR-195 to regulate PC/ISC niche function by altering PHB1 expression and mitochondrial metabolism. These findings provide better understanding of the mechanisms underlying the maintenance of intestinal homeostasis in stressful environments and point to potential therapeutic targets to enhance regeneration and adaptation of the intestinal mucosa in surgical patients with critical disorders.

# Materials and Methods

### Studies in murine and human tissues

Age- and gender-matched 6- to 8-wk-old mice of C57BL/6 background were used. Intestinal epithelial tissue-specific HuR deletion (IE-HuR$^{-/-}$) mice were generated by crossing the HuR$^{flox/flox}$ (HuR$^{fl/fl}$) and villin-Cre mice purchased from the Jackson Laboratory, as described in our previous studies (Liu et al, 2014; Xiao et al, 2019). HuR$^{fl/fl}$-Cre$^-$ mice served as control littermates. Both IE-HuR$^{-/-}$ mice and control littermates were housed and handled in a pathogen-free breeding barrier and were cared for by trained technicians and veterinarians. Animals were deprived of food but were allowed free access to tap water for 24 h before experiments. Two portions of the middle small intestine were taken, one for histological examination and the other for extraction of protein and RNA. The tissues were fixed in formalin and paraffin for immunohistochemical staining, whereas the mucosa was scraped with a glass slide for various measurements, as described previously (Yu et al, 2011; Xiao et al, 2018). All animal experiments were performed in accordance with NIH guidelines and were approved by the Institutional Animal Care and Use Committee of University of Maryland School of Medicine and Baltimore VA Hospital.

Human tissue samples were obtained from surplus discarded tissue from the University of Maryland Health Science Center and Penn State Hershey Carlino Family Inflammatory Bowel and Colorectal Disease Biobank. All patients gave informed consent to have surgically resected tissue collected for this study. The diseased intestinal tissues, as determined by a pathologist and surgeon, were stored in liquid nitrogen until they were assayed. Dissected and opened intestines were mounted onto a solid surface and fixed in formalin and paraffin. The study was approved by the Institutional Review Boards of University Maryland and the Pennsylvania State College of Medicine.

### Cell and intestinal organoid culture

Human colorectal carcinoma Caco-2 cells were purchased from the American Type Culture Collection and were maintained under

standard culture conditions (Liu et al, 2017; Xiao et al, 2018). The culture medium and fetal bovine serum were purchased from Invitrogen and biochemical reagents were from Sigma-Aldrich. Isolation and culture of primary enterocytes were conducted following the method described previously (Yu et al, 2020b; Yu et al, 2022). Briefly, primary crypts were released from the small intestinal mucosa of mice; isolated crypts were mixed with Matrigel (Corning) and cultured in mouse IntestiCult organoid growth medium (Stemcell technology). The levels of DNA synthesis were measured by assaying BrdU incorporation, and the growth of organoids was examined by measuring organoid cross-sections using NIS-Elements AR4.30.02 program.

## Plasmid construction and RNA interference

An expression vector containing the human HuR cDNA under the control of pCMV promoter was purchased from Origene and used to increase cellular HuR levels as described previously (Liu et al, 2009). Transient transfections were performed using the Lipofectamine reagent following the manufacturer's recommendations (Invitrogen). 48 h after transfection using LipofectAMINE, cells were harvested for analysis. Expression of HuR was silenced by transfection with siHuR as described (Liu et al, 2017). The siHuR and C-siRNA were purchased from Santa Cruz Biotechnologies. For each 60-mm cell culture dish, 15 $\mu$l of the 20 $\mu$M stock duplex siHuR or C-siRNA was used. 48 h after transfection using LipofectAMINE (116668019; Invitrogen), the cells were harvested for analysis.

## Q-PCR and immunoblotting analyses

Total RNA was isolated by using the RNeasy mini kit (QIAGEN) and used in RT and quantitative (Q)-PCR amplification reactions as described (Zhuang et al, 2013; Yu et al, 2022). Q-PCR analysis was performed using Step-one-plus Systems with specific primers, probes, and software (Applied Biosystems). All primers used for Q-PCR analysis were purchased from Thermo Fisher Scientific. The levels of *Gapdh* mRNA were assessed to monitor the evenness in RNA input in Q-PCR analysis.

To examine protein levels, whole-cell lysates were prepared using 2% SDS, sonicated, and centrifuged (Xiao et al, 2016). The supernatants were boiled and size-fractionated by SDS–PAGE. After transferring proteins onto nitrocellulose filters, the blots were incubated with primary antibody, after incubations with secondary antibody. Antibodies recognizing HuR, PHB1, PHB2, HSP60, SGK, VDAC, Cyto-C, PDHG, and GAPDH were obtained from Santa Cruz Biotechnology and BD Biosciences and Invitrogen. Secondary antibodies conjugated to horseradish peroxidase were purchased from Sigma-Aldrich. All antibodies used in this study were validated for species specificity. Antibody dilutions used for Western blots to detect HuR, PHB1, PHB2, HSP60, SGK, VDAC, Cytochrome C, Lgr5, PDHG, and GAPDH were 1:800 or 1,000 (first Ab) and 1:2,000 (second Ab), respectively, whereas antibody dilutions for immunostaining were 1:200 (first Ab) and 1:2,000 (second Ab). Relative protein levels were analyzed by using Bio-Rad Chemidoc and XRS system equipped with Image Lab Software (version 4.1). We also used "Quantity tool" to determine the band intensity volume; the values were normalized with internal loading control GAPDH.

## Immunofluorescence staining and TEM

The procedures of immunofluorescence staining were carried out according to the method described (Xiao et al, 2019; Yu et al, 2020b). Slides were fixed in 3.7% formaldehyde in PBS and rehydrated. All slides were incubated with a primary antibody against different proteins in the blocking buffer at concentration of 1:200 or 1:300 dilution at 4°C overnight and then incubated with a secondary antibody conjugated with Alexa Fluor-594 (Invitrogen) or Alexa Fluor-488 (Invitrogen) for 2 h at RT. After rinsing three times, the slides were incubated with 1 $\mu$M DAPI (Invitrogen) for 10 min to stain cell nuclei. Finally, the slides were mounted and viewed through a Zeiss confocal microscope (LSM710; model). Slides were examined in a blinded fashion by coding, and decoded only after examination was completed. Images were processed using Photoshop software (Adobe).

For TEM, the small intestines from the animals were rapidly removed, cut into small pieces, and placed into 2% paraformaldehyde and 2.5% glutaraldehyde in 0.1 M sodium cacodylate buffer for 1 h. Samples were incubated in 1% osmium tetroxide for 1 h and then dehydrated through an ethanol series. They were embedded in EPON resin and ultrathin sections of 80–100 nm were collected on grids. The samples were imaged in a FEI Tecnai T12 TEM (Thermo Fisher Scientific) at 80 kV with an XR60B AMT CCD camera (2 × 2 k).

## Isolation of IECs and scRNA-seq analysis

The small intestinal mucosal tissues harvested from littermate and IE-HuR$^{-/-}$ mice were weighed before being washed in cold D-PBS and diced with scalpel, as described previously (Xiao et al, 2016). Briefly, cleaned mucosal tissues were incubated in chelation medium (20 mM; EDTA-DPBS) at 37°C for 90 min with agitation. Villus-rich supernatant was collected and later combined with a pellet that was further dissociated with TrypLE for 10 min at 37°C (crypt-rich cells). The epithelial single-cell suspension was then washed and passed through 40-$\mu$m filters. Before proceeding to scRNA-seq, the purity of epithelial populations was confirmed by flow cytometry analysis. Once satisfactory viability and EPCAM purity were demonstrated, these high-quality IECs were directly loaded for droplet-based scRNA-seq according to the manufacturer's protocol for the Chromium Single Cell Platfor (3′V2; 10X Genomics) to obtain 10,000 cells per reaction. Library preparation was carried out according to the manufacturer's protocol. 10× Genomics scRNA-seq gene expression raw sequencing data were processed using the CellRanger software v.3.0.2 and the 10X human transcriptome GRCh38-3.0.0 as the reference. The 10X Genomics V (D) Ig heavy and light chains were processed using Cellranger vdj v.3.1.0 and the reference Cellranger-vdj-GRCh38-alts-ensembl-3.1.0 with default settings. The dimensionality reduction, Leiden clustering, differential abundance analysis, cell-type composition analysis, gene enrichment analysis, mitochondrion Gene Ontology Term (GO: 0005739) were performed as reported previously (Islam et al, 2020; Elmentaite et al, 2021).

## Seahorse metabolic analyzer assays

Seahorse Bioscience XFe24 Analyzer was used to measure mitochondrial respiratory capacity in intestinal organoids and cultured

IECs (Thompson et al, 2019; Kleele et al, 2021). 2-D intestinal organoids and Caco-2 cells were grown on a 96-well XFe96 plate at a cell density of 20,000 cells/well. Cartridge plates for metabolic stress injections were hydrated for at least 12 h at 37°C before the assay with Calibrant Solution. 1 h before running the Seahorse assay, the cell culture medium was removed and replaced with Seahorse Assay Medium. The following compounds (final concentrations) were sequentially injected into each well: oligomycin (1.5 µM), FCCP (0.25 µM), and rotenone/antimycin (0.5 µM). Oxygen consumption rate was measured under basal conditions and after each injection using an XFe96 extracellular flux analyzer (Seahorse Bioscience). Key parameters of mitochondrial function, including basal respiration, ATP-linked respiration, proton leak, maximal respiration and spare capacity, were calculated and analyzed on Wave (Agilent). Mitochondrial activity and intracellular superoxide production were examined by using MitoTracker Red and MitoSox kits (Invitrogen) and performed, according to the manufacturer's instruction.

### RNP-IP and biotin-labeled miR-195 pull-down assays

Immunoprecipitation (IP) of RNP complexes was carried out to assess the association of endogenous HuR with endogenous mRNAs encoding PHB1, PHB2, HSP60, and claudin-1 as described (Liu et al, 2009). Twenty million cells were collected per sample, and lysates were used for IP for 4 h at RT in the presence of excess (30 µg) IP antibody (IgG, or anti-HuR). RNA in IP materials was used in RT reactions followed by Q-PCR analysis. The amplification of *Gapdh* mRNA, found in all samples as low-level contaminating housekeeping transcripts (not HuR target), served to monitor the evenness of sample input, as reported previously (Zhuang et al, 2013).

Biotinylated RNA pull-down assays were conducted as described previously (Kwon et al, 2021). After biotin-labeled miR-195 was incubated with cytoplasmic proteins in RT for 1 h, the mixture was mixed with Streptavidin–Dynabeads (Invitrogen) and incubated at 4°C on a rotator overnight. The beads were washed thoroughly, and the beads-bound RNA was isolated and subjected to RT followed by Q-PCR analysis.

### Statistical analysis

All values were expressed as the means ± SEM. Unpaired, two-tailed *t* test was used when indicated with $P < 0.05$ considered significant. When assessing multiple groups, one-way ANOVA was used with Tukey's post hoc test (Harter, 1960). The statistical software used was GraphPad InStat Prism 9.0. For nonparametric analysis rank comparison, the Kruskal–Wallis test was conducted.

## Data Availability

Primary datasets of single-cell transcriptional profiles of PCs and ISCs in IE-HuR$^{-/-}$ and littermate mice have been generated and deposited on NCBI with GEO Accession number GSE242410.

## Supplementary Information

## Acknowledgements

This work was supported by Merit Review Awards (to J-Y Wang and JN Rao) from US Department of Veterans Affairs; grants from National Institutes of Health (NIH) (DK57819, DK61972, DK68491 to J-Y Wang); and funding from the National Institute on Aging-Intramural Research Program, NIH (to M Gorospe) and from the Peter and Marshia Carlino Fund for IBD Research (to GS Yochum and WA Koltun).

### Author Contributions

L Xiao: data curation, formal analysis, validation, and methodology.
B Warner: data curation.
CG Mallard: data curation.
HK Chung: data curation.
A Shetty: software, validation, and methodology.
CA Brantner: data curation and methodology.
JN Rao: data curation and supervision.
GS Yochum: data curation.
WA Koltun: data curation.
KB To: investigation.
DJ Turner: data curation and methodology.
M Gorospe: investigation.
J-Y Wang: conceptualization, supervision, funding acquisition, investigation, methodology, project administration, and writing—original draft, review, and editing.

### Conflict of Interest Statement

The author discloses the following: J-Y Wang is a Senior Research Career Scientist at the Biomedical Laboratory Research and Development Service (US Department of Veterans Affairs). The remaining authors disclose no conflicts of interest.

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
