## [Reviewer comments · Life Science Alliance]

Life Science Alliance

Control of Paneth cell function by HuR regulates gut mucosal growth by altering stem cell activity

Lan Xiao, Bridgette Warner, Caroline Mallard, Hee Chung, Amol Shetty, Christine Brantner, Jaladanki Rao, Gregory Yochum, Walter Koltun, Kathleen To, Douglas Turner, Myriam Gorospe, and Jian-Ying Wang

DOI: <https://doi.org/10.26508/lsa.202302152>

Corresponding author(s): *Jian-Ying Wang, University of Maryland, Baltimore*

Review Timeline:

Submission Date:	2023-05-12
Editorial Decision:	2023-06-12
Revision Received:	2023-08-07
Editorial Decision:	2023-08-23
Revision Received:	2023-08-29
Accepted:	2023-08-30

Scientific Editor: Novella Guidi

Transaction Report:

June 12, 2023

Re: Life Science Alliance manuscript #LSA-2023-02152-T

Prof. Jian-Ying Wang
University of Maryland
Departments of Surgery and Pathology
10 N. Greene Street
Baltimore, MD 21201

Dear Dr. Wang,

Thank you for submitting your manuscript entitled "Control of Paneth cell function by HuR regulates gut mucosal growth by altering stem cell activity" to Life Science Alliance. The manuscript was assessed by expert reviewers, whose comments are appended to this letter. We invite you to submit a revised manuscript addressing the Reviewer comments. The point raised by reviewer 3 about the need to use lineage-specific Cre mice (Lgr5-Cre and Lyz-Cre) can be overruled.

Thank you for this interesting contribution to Life Science Alliance. We are looking forward to receiving your revised manuscript.

Sincerely,

B. MANUSCRIPT ORGANIZATION AND FORMATTING:

Reviewer #1 (Comments to the Authors (Required)):

Integrity and effectiveness of intestinal stem cells (ISCs) and Paneth cells (PCs) niche are essential for constant renewal of the intestinal epithelium. The current study by Lan Xiao et al. aimed to prove that control of Paneth cell function by HuR via mitochondria regulates renewal of the intestinal epithelium by altering stem cell activity. Targeted deletion of HuR in mice disrupted PC gene expression profiles, reduced PC-derived niche factors, and impaired ISC function, leading to inhibited renewal of the intestinal epithelium. Human intestinal mucosa from patients with critical surgical disorders exhibited decreased levels of tissue HuR and PC/ISC niche dysfunction, along with disrupted mucosal growth. HuR deletion led to mitochondrial impairment by decreasing the levels of several mitochondrial-associated proteins including prohibitin 1 (PHB1) in the intestinal epithelium, whereas HuR enhanced PHB1 expression by preventing miR-195 binding to the Phb1 mRNA. These results indicate that HuR is essential for maintaining the integrity of the PC/ISC niche and highlight a novel role for a defective PC/ISC niche in the pathogenesis of intestinal mucosa atrophy. The current findings provide better understanding of the mechanisms underlying the maintenance of intestinal homeostasis in stressful environments and point to potential therapeutic targets to enhance regeneration and adaptation of the intestinal mucosa in surgical patients with critical disorders. The study design of the experiments is rigorous, and the data presented in the manuscript is very strong and supports the authors' central hypothesis.

Minor points:

1. Recent studies have demonstrated that the protein level of intact HuR was significantly decreased, whereas cleaved HuR (HuR CP-1) was significantly increased, in the colon of an IBD mouse model. This finding is important because the cleaved product (approximately 24 kDa) was earlier shown to change the binding ability of HuR to occludin and E-cadherin mRNA 3' untranslated regions (UTRs). The related findings should be verified in the isolated colon tissues from IBD patients and IE-HuR^{-/-} mice in future studies.
2. Besides PHB1, miR-195-5p also regulates tight junction expression via Claudin-2 downregulation in Ulcerative Colitis. The authors showed that the Claudin-1 (Cldn1) mRNA was highly enriched in HuR samples compared with control IgG. The role of Claudin-2 should be briefly discussed in the manuscript.
3. Paneth cells as highly susceptible to mitochondrial dysfunction and central to the pathogenesis of ileitis, with translational implications for the subset of Crohn's disease patients exhibiting Paneth cell defects. Therefore, the authors may want to consider verifying the related Paneth cell defects in the 4 Crohn's disease (CD) patients along with the normal controls.
4. In Fig. 5, besides oxygen consumption rate, the related metabolic differentiation and oxygen metabolism induced by HuR silencing should also be discussed.

Reviewer #2 (Comments to the Authors (Required)):

The manuscript "Control of Paneth cell function by HuR via mitochondria regulates renewal of the intestinal epithelium by altering stem cell activity" presents interesting findings on the relationship between HuR and PHB1 in regulating Paneth cells and stem cell activity, which impacts gastrointestinal research. Overall, this work is well designed and carefully performed. However, there are several concerns that the authors should address:

- (1) The main phenotype observed in HuR IEC-specific knockout mice is the deficiency in Paneth cells and stem cell activity. Although the authors provide mechanistic insights into the regulation of mitochondrial dysfunction through PHB1 expression, the majority of the mechanistic studies were conducted in the Caco-2 cell line (a human colorectal adenocarcinoma derived cell line), which may not accurately represent the molecular mechanisms involved in small intestinal Paneth cells and stem cells. Authors need to discuss this limitation in the manuscript.
- (2) In Figure 1, the authors conclude that HuR IEC knockout mice exhibit decreased cell proliferation based on observations of crypt and villi shrinkage. To strengthen this conclusion, it would be great to confirm the decrease in cell proliferation by staining

with proliferation markers such as Mki-67 or PCNA or by using in vivo labeling with BrdU or EdU approach. Alternatively, authors could discuss this limitation in the paper.

(3) It is wonder if the authors can investigate the impact of HuR IEC knockout on secretory progenitor cells (Atoh1+) and Goblet cells, provide relevant data, or discuss this aspect.

(4) Regarding the single-cell RNA sequencing data in Figure 2B, 2C, and Figure 4A, the authors should specify the p-value cutoff used for identifying differentially expressed genes (DEGs) and conducting pathway analysis.

(5) The authors report that HuR IEC knockout affects the expression of ISC/progenitor cell markers based on single-cell RNA sequencing data. However, typical ISC markers such as Lgr5, Olfm4, and Ascl2 are not among the top DEGs listed in Figure 2C. Additional verification through qPCR is recommended to validate the single-cell RNA sequencing data.

(6) Careful examination of the scale bars is necessary as Figure 1E and Figure 3B appear to have incorrect scale bar representation.

(7) The y-axis is out-of-range in Figure 1F, Figure 5E, Supplementary Figure 1D, and Supplementary Figure 3A. Adjustments should be made to ensure clear visualization of the data.

(8) The authors should deposit the RNA sequencing data and provide the submission information upon revising the manuscript. They also have the right to set the data release date.

Reviewer #3 (Comments to the Authors (Required)):

This is an interesting (though mainly descriptive and associative) study on the putative role of HuR in regulating self-renewal and regenerative response in the (small) intestinal stem cell niche.

The main problem I encountered is that the authors did not establish a specific cause-effect relationship between loss of HuR in a specific intestinal lineage but rather used tools (villinCre) that are common to virtually all cell types of the crypt of Lieberkühn. The first main and unanswered question is: where along the crypt-villus axis HuR is expressed. This should be relatively easy to assess not only by IF but also taking advantage of the scRNAseq data and the corresponding UMAP analyses. In particular, it is of quintessential importance to determine whether Paneth cells and/or Lgr5+ ISCs do express the gene in question. This is essential to determine whether the observed defects are caused by a direct effect of loss of HuR in ISCs, or does it work through Paneth cells and their well-known niche role? The only way around this is to use lineage-specific Cre mice (Lgr5-Cre and Lyz-Cre) to induce the deletion and, based on the phenotypic effects, identify the cellular culprit. The same is true for the organoids: as long as these are derived from resected whole crypts, the authors will not be able to identify the responsible cell lineage. Instead, elegant reconstitution assays can be implemented by FACSorting Paneth cells and ISCs from both control and ablated mice and co-incubate in all possible combinations followed by a count of organoid multiplicity and analysis of their morphology.

Some indications may also arise from the analysis of mitochondrial gene activity. It has been established that while ISCs have a pronounced mitochondrial metabolism, Paneth cells are prevalently glycolytic. Yet, the demonstrated defects are shown in what the authors refer to as Paneth-like cells (?). In EM, Paneth cells are clearly recognizable for the presence of secretory granules and endoplasmatic reticulum, while the slender Lgr5 ISCs are squeezed in between two PCs. Personally, I do not see either one in the presented EM figures.

To sum up, the authors should more specifically address the HuR expressing lineage that is causing the defect and which are the underlying molecular and cellular mechanisms.

One minor point relates to the specificity of the Lgr5 Ab's used for the identification of ISCs. To my knowledge, such Ab's (with proven specificity) are unfortunately not yet available and that is why most labs use Lgr5-GFP knock-in mice. In some of the IFs, Lgr5+ cells appear to map high in the crypt-villus axis which is by no means normal.

Reply to Reviewer A

We greatly appreciate the time and effort spent to help us improve this manuscript. We have addressed all your concerns as thoroughly as possible.

Question 1: *Recent studies have demonstrated that the protein level of intact HuR was significantly decreased, whereas cleaved HuR (HuR CP-1) was significantly increased, in the colon of an IBD mouse model. This finding is important because the cleaved product (approximately 24 kDa) was earlier shown to change the binding ability of HuR to occludin and E-cadherin mRNA 3' untranslated regions (UTRs). The related findings should be verified in the isolated colon tissues from IBD patients and IE-HuR^{-/-} mice in future studies.*

Response: We thank the reviewer for this specific comment. It is true that cleaved HuR (HuR CP-1) increased in IBD patients and mouse models, as reported recently, but the function and impact of altered levels of cleaved HuR in the pathogenesis in IBD remain unknown. As pointed out by the reviewer and the results from our previous studies (EMBO J/2009; Mol Cell Biol/2018), cleaved HuR, particularly from ubiquitin-mediated HuR proteolysis, fails to bind to target mRNAs including the transcripts encoding occludin and E-cadherin. Induction in the level of cleaved HuR in the colon of IBD suggests that decreased abundance of intact HuR in IBD might result from HuR degradation, but this increase in cleaved HuR may have only limited biological function in the intestinal epithelium. In addition, it is impossible to verify changes in the levels of cleaved HuR in the intestinal mucosa after HuR deletion in mice. As shown in Figures 1 and 4, HuR in the intestinal mucosa was undetectable in IE-HuR^{-/-} mice, because the *HuR* gene was deleted from the mouse genome in this study. We hope the reviewer would agree with us that the exact biological function of cleaved HuR in the intestinal epithelium homeostasis and its role in the pathogenesis of IBD should be fully investigated in a separate study in future.

Question 2: *Besides PHB1, miR-195-5p also regulates tight junction expression via Claudin-2 downregulation in Ulcerative Colitis. The authors showed that the Claudin-1 (Cldn1) mRNA was highly enriched in HuR samples compared with control IgG. The role of Claudin-2 should be briefly discussed in the manuscript.*

Response: Yes, miR-195 also down-regulates claudin 2, although the basal level of claudin-2 in the intestinal epithelium is extremely low compared with those of claudin-1 and E-cadherin. It is possible that altered claudin-2 level by HuR deletion also contributes to pathological changes of the intestinal epithelium in IE-HuR^{-/-} mice. The potential impact of claudin 2 in HuR-modulated intestinal mucosal homeostasis is briefly discussed in the revised manuscript (pages 17, and 18).

Question 3: *Paneth cells as highly susceptible to mitochondrial dysfunction and central to the pathogenesis of ileitis, with translational implications for the subset of Crohn's disease patients exhibiting Paneth cell defects. Therefore, the authors my want to consider verifying the related Paneth cell defects in the 4 Crohn's disease (CD) patients along with the normal controls.*

Response: In this study, we examined the clinical relevance of HuR function in the human Paneth cells (PCs) and function of PC/intestinal stem cell (ISC) niche in human ileal mucosal tissues collected from four Crohn's disease (CD) patients who required urgent/emergent intestinal resection due to severe complications (SC) and four healthy controls who had no CD nor emergency surgical disorders. Consistent with our previous findings observed in CD patients without SC (Xiao *et al.*, Gastroenterology/2019), the ileal mucosa from patients with CD/SC also exhibited significant defects in PCs, along with an inactivation of PC/ISC nice function. These results are presented in Figure 3 and described in the text of revised manuscript (page 9).

Question 4: Besides oxygen consumption rate, the related metabolic differentiation and oxygen metabolism induced by HuR silencing should also be discussed.

Response: We greatly appreciate the reviewer for the specific comment. In this study, primary cultured intestinal organoids and cultured IECs were used to examine mitochondrial function. Our results show that HuR deletion caused significant decreases in basal and maximal respiration levels, ATP production, and spare respiratory capacity in both organoids and IECs. These exciting findings are provided in Figure 5 and discussed in the revised manuscript (pages 12 and 13).

Reply to Reviewer B

We greatly appreciate the constructive questions and positive comments provided by this Referee.

Question 1: The main phenotype observed in HuR IEC-specific knockout mice is the deficiency in Paneth cells and stem cell activity. Although the authors provide mechanistic insights into the regulation of mitochondrial dysfunction through PHB1 expression, the majority of the mechanistic studies were conducted in the Caco-2 cell line (a human colorectal adenocarcinoma derived cell line), which may not accurately represent the molecular mechanisms involved in small intestinal Paneth cells and stem cells. Authors need to discuss this limitation in the manuscript.

Response: We agree with the reviewer that Caco-2 cells used as an *in vitro* model in this study have a limitation and that the results regarding mechanistic studies obtained from Caco-2 cells should be verified in *ex vivo* and *in vivo* systems in the future. Currently, stable lines of Paneth cells, intestinal stem cells, and secretory (differentiated) cells are not available, and Caco-2 cells are commonly used as an *in vitro* model studying intestinal epithelial cell biology. Although IE-HuR^{-/-} mice, primarily cultured organoids, human mucosal tissues, and cultured CIECs were used in this study, we pointed out the limitation of mechanistic studies conducted in Caco-2 cells and appropriately revised our conclusions in the revised manuscript (page 17).

Question 2: In Figure 1, the authors conclude that HuR IEC knockout mice exhibit decreased cell proliferation based on observations of crypt and villi shrinkage. To strengthen this conclusion, it would be great to confirm the decrease in cell proliferation by staining with proliferation markers such as Mki-67 or PCNA or by using *in vivo* labeling with BrdU or EdU approach. Alternatively, authors could discuss this limitation in the paper.

Response: Yes, we did examine cell proliferation in the intestinal mucosa by using BrdU staining and demonstrated that HuR deletion decreased the numbers of BrdU-positive cells within the crypts of small intestine but not in colonic mucosa. The inhibition of small intestinal mucosal growth by HuR deletion in IE-HuR^{-/-} mice was similar to those reported in our previous studies (Liu et al., 2014; 2017). This information is provided in the revised manuscript (page 6).

Question 3: It is wonder if the authors can investigate the impact of HuR IEC knockout on secretory progenitor cells (Atoh1+) and Goblet cells, provide relevant data, or discuss this aspect.

Response: Both secretory progenitor cells (Atoh1+) and Goblet cells in the intestinal epithelium of IE-HuR^{-/-} and littermate mice were clustered in our database of scRNA-seq analysis. We will dissect the data for each cluster and analyze the impact of HuR on their transcriptional regulation and function in the future study. In this study, we highly focused on the role of HuR in the

regulation of Paneth cell function but briefly discuss potential impact of HuR in secretory progenitor and Goblet cells in the revised manuscript (page 8).

Question 4: *Regarding the single-cell RNA sequencing data in Figure 2B, 2C, and Figure 4A, the authors should specify the p-value cutoff used for identifying differentially expressed genes (DEGs) and conducting pathway analysis.*

Response: We thank the reviewer for this specific comment. The p-value cut-off used for identifying differentially expressed genes was 0.05. This information is provided in Figure Legends of the revised manuscript (pages 31, 32).

Question 5: *The authors report that HuR IEC knockout affects the expression of ISC/progenitor cell markers based on single-cell RNA sequencing data. However, typical ISC markers such as Lgr5, Olfm4, and Ascl2 are not among the top DEGs listed in Figure 2C. Additional verification through qPCR is recommended to validate the single-cell RNA sequencing data.*

Response: In analysis of our data from scRNA-seq analysis, Paneth cells were marked by high expression of *Defe5* and *Rg3a* mRNAs, while intestinal stem cells were marked by transcripts including *Ascl2*, *Lgr5*, *Olfm4*, *Rgmb*, and *Smoc2* mRNAs, as reported previously (Elmentaite et al, 2021; Luna Velez et al, 2023). The top DEGs listed in Figure 2B and C are genes that are involved in functions of Paneth cells and ISCs, respectively, while expression levels of some these cell marker genes are not included in the lists. We agree with the reviewer that immunostaining assays or qPCR should be used to validate the scRNA-seq data in the future study. This information is provided in the revised manuscript (page 8).

Question 6: *Careful examination of the scale bars is necessary as Figure 1E and Figure 3B appear to have incorrect scale bar representation.*

Response: We thank the reviewer for pointing out the errors and we have made the corrections in Figure 1E and Figure 3B, respectively.

Question 7: *The y-axis is out-of-range in Figure 1F, Figure 5E, Supplementary Figure 1D, and Supplementary Figure 3A. Adjustments should be made to ensure clear visualization of the data.*

Response: We thank the reviewer for pointing out the possible errors in our figures. We have re-examined the data in Figure 1F and 5E, supplementary 1D and 3A, and made sure that there is no data out-of-range.

Question 8: *The authors should deposit the RNA sequencing data and provide the submission information upon revising the manuscript. They also have the right to set the data release date.*

Response: We appreciate the reminder from the reviewer. All our RNA sequencing data will be provided with the publication following the journal's guideline.

Reply to Referee #3

We greatly appreciate the time and effort spent to help us improve this manuscript. We believe that we have now thoroughly addressed all your remaining concerns.

Question: *The main problem I encountered is that the authors did not establish a specific cause effect relationship between loss of HuR in a specific intestinal lineage but rather used tools (villinCre) that are common to virtually all cell types of the crypt of Lieberkühn. The first main and unanswered question is: where along the crypt-villus axis*

HuR is expressed. This should be relatively easy to assess not only by IF but also taking advantage of the scRNAseq data and the corresponding UMAP analyses. In particular, it is of quintessential importance to determine whether Paneth cells and/or Lgr5+ ISCs do express the gene in question. This is essential to determine whether the observed defects are caused by a direct effect of loss of HuR in ISCs, or does it work through Paneth cells and their well-known niche role? The only way around this is to use lineage-specific Cre mice (Lgr5-Cre and Lyz-Cre) to induce the deletion and, based on the phenotypic effects, identify the cellular culprit. The same is true for the organoids: as long as these are derived from resected whole crypts, the authors will not be able to identify the responsible cell lineage. Instead, elegant reconstitution assays can be implemented by FAC Sorting Paneth cells and ISCs from both control and ablated mice and co-incubate in all possible combinations followed by a count of organoid multiplicity and analysis of their morphology.

Response: We appreciate the concerns raised by this reviewer. Based on our previous studies and others, HuR is highly expressed in all crypt cells of the small intestinal mucosa, including Paneth cells and stem cells. Experiments using immunostaining and Western immunoblotting analysis (using proteins from different cellular fractions) further showed that HuR is predominantly distributed in the nucleus in unstimulated intestinal epithelial cells. In this study, we found that targeted deletion of HuR in all intestinal epithelial cells disrupted Paneth cell gene expression profiles, reduced Paneth cell-derived niche factors, and impaired intestinal stem cell function, leading to inhibited renewal of the intestinal epithelium. We agree with the reviewer that crossbreeding of HuR^{flox/flox} mice with lineage-specific Cre mice (Lgr5-Cre and Lyz-Cre) will provide a powerful tool to induce Paneth cell- or stem cell-specific HuR deletion in the intestinal epithelium and these new animal models will improve the quality of our study. However, generating and characterizing new cell type-specific mouse models will take years, and we like to perform these experiments in our future study.

Question: *Some indications may also arise from the analysis of mitochondrial gene activity. It has been established that while ISCs have a pronounced mitochondrial metabolism, Paneth cells are prevalently glycolytic. Yet, the demonstrated defects are shown in what the authors refer to as Paneth-like cells (?). In EM, Paneth cells are clearly recognizable for the presence of secretory granules and endoplasmatic reticulum, while the slender Lgr5 ISCs are squeezed in between two PCs. Personally, I do not see either one in the presented EM figures.*

Response: To test the possibility that HuR regulates activity of the Paneth cells via mitochondria metabolism, we screened for expression of mitochondrial-associated genes in our single-cell sequencing data from Paneth cells (marked by *Defe5* and *Rg3a*). Our results show that HuR deletion decreased the expression levels of many mRNAs encoding proteins that control mitochondrial catabolism and oxidation/reduction reactions. It is true that typical Paneth cells commonly have secretory granules (SGs) and endoplasmatic reticulum (ER), but they are not specific cell markers for Paneth cells and the levels of cellular SGs and ER are affected by many factors such as stress, food starvation, and different methods for tissue preparation. In our study, both littermate and IE-HuR^{-/-} mice were fasted for 24 h before experiments, and the standard protocol for tissue fixation and slide preparation was used as reported by others. We did not see any SGs and ER in the entire crypt areas of both control littermate and IE-HuR^{-/-} mice under EM. Although Paneth cells cannot be fully confirmed and identified in our EM images, we defined Paneth-like cells based on **a**) their location in crypt; **b**) morphology; and **c**) enrichment of mitochondria with the assistance of a pathologist. Our results showed that mitochondria in PC-like cells exhibited swollen morphology, disruption of cristae, decreased fused structures, and occasional dense inclusion bodies in IE-HuR^{-/-} mice, compared with those observed in littermate

mice (Figure 4D). On the other hand, HuR deletion did not alter the morphology and number of mitochondria in enterocytes located at the villous area of the small intestinal mucosa (Supplementary Figure 3B).

Question: *One minor point relates to the specificity of the Lgr5 Ab's used for the identification of ISCs. To my knowledge, such Ab's (with proven specificity) are unfortunately not yet available and that is why most labs use Lgr5-GFP knock-in mice. In some of the IFs, Lgr5+ cells appear to map high in the crypt-villus axis which is by no means normal.*

Response: Antibody against Lgr5 was purchased from Invitrogen (Cat: MA5-25644) and diluted at 1:200 for immunostaining assay. This antibody works well with a high specificity in our study.

August 23, 2023

RE: Life Science Alliance Manuscript #LSA-2023-02152-TR

Prof. Jian-Ying Wang
University of Maryland, Baltimore
Departments of Surgery and Pathology
10 N. Greene Street
Baltimore, MD 21201

Dear Dr. Wang,

Thank you for submitting your revised manuscript entitled "Control of Paneth cell function by HuR regulates gut mucosal growth by altering stem cell activity". We would be happy to publish your paper in Life Science Alliance pending final revisions necessary to meet our formatting guidelines.

- please upload all figure files as individual ones, including the supplementary figure files; all figure legends should only appear in the main manuscript file
- please add ORCID ID for the corresponding author--you should have received instructions on how to do so
- please add a Summary Blurb/Alternate Abstract to our system
- please add a Category and keywords for your manuscript to our system
- please add the Twitter handle of your host institute/organization as well as your own or/and one of the authors in our system
- please note that titles in the system and on the manuscript file must match
- please remove header "Ms. #LSA-2023-02152-T, R1 Revised: 7-29-2023" from your manuscript file
- please add your main and supplementary figure legends to the main manuscript text after the references section
- please add callouts for Figure S1A-D to your main manuscript text;
- please add the deposition number and in which public database your single-cell transcriptional profiles were deposited

Figure checks:

- please add molecular weights next to blots in Fig 4B, 5C, 6F, 6H, and S4B
- there is a horizontal line in blots in Figure 4B Cyto-C. Please provide source data.
- there is a splice in figure 6F, the second blot -PHB2. Please provide source data.

A. FINAL FILES:

B. MANUSCRIPT ORGANIZATION AND FORMATTING:

Sincerely,

Reviewer #1 (Comments to the Authors (Required)):

The authors have done an excellent job of responding to all of my points.

Reviewer #2 (Comments to the Authors (Required)):

In this revised version, the authors have adeptly addressed the comments provided during the previous review. The findings put forth are both novel and captivating, and the execution of the work is commendable. No further concerns have arisen. This reviewer holds a strong enthusiasm and recommends the acceptance of this manuscript for publication.

August 30, 2023

RE: Life Science Alliance Manuscript #LSA-2023-02152-TRR

Prof. Jian-Ying Wang
University of Maryland, Baltimore
Departments of Surgery and Pathology
10 N. Greene Street
Baltimore, MD 21201

Dear Dr. Wang,

Thank you for submitting your Research Article entitled "Control of Paneth cell function by HuR regulates gut mucosal growth by altering stem cell activity". It is a pleasure to let you know that your manuscript is now accepted for publication in Life Science Alliance. Congratulations on this interesting work.

DISTRIBUTION OF MATERIALS:

Again, congratulations on a very nice paper. I hope you found the review process to be constructive and are pleased with how the manuscript was handled editorially. We look forward to future exciting submissions from your lab.

Sincerely,
